# Influence of Various Cooking Methods on Selenium Concentrations in Commonly Consumed Seafood Species in Thailand

**DOI:** 10.3390/foods14152700

**Published:** 2025-07-31

**Authors:** Narisa Rueangsri, Kunchit Judprasong, Piyanut Sridonpai, Nunnapus Laitip, Jörg Feldmann, Alongkote Singhato

**Affiliations:** 1Nutrition and Dietetics Division, Faculty of Allied Health Sciences, Burapha University, Mueng Chonburi 20131, Thailand; narisar@go.buu.ac.th; 2Institute of Nutrition, Mahidol University, Salaya, Phutthamonthon, Nakhon Pathom 73170, Thailand; kunchit.jud@mahidol.ac.th (K.J.); piyanut.sri@mahidol.ac.th (P.S.); 3Inorganic Analysis Group, Chemical Metrology and Biometry Department, National Institute of Metrology (Thailand), Pathum Thani 12120, Thailand; nunnapusl@nimt.or.th; 4Trace Element Speciation Laboratory (TESLA), Institute of Chemistry, University of Graz, 8010 Graz, Austria; joerg.feldmann@uni-graz.at

**Keywords:** true retention, trace elements, food analysis, seafood, selenium

## Abstract

Selenium (Se) is an important trace element in our body; however, food composition data remain limited due to analytical challenges and interferences. Seafood, abundant in Thailand, is recognized as a rich source of Se. This study aimed to expand knowledge on Se content in seafood prepared using traditional Thai cooking methods. Twenty seafood species were selected and prepared by boiling, frying, and grilling. Inductively Coupled Plasma–Triple Quadrupole–Mass Spectrometry (ICP-MS/MS) was used to analyze total Se contents in selected seafood species. Results revealed significant variation in Se content across species and cooking methods. The Indo-Pacific horseshoe crab showed the highest Se concentration, with fried samples reaching 193.9 μg/100 g. Se concentrations were in the range of 8.6–155.5 μg/100 g (fresh), 14.3–106.6 μg/100 g (boiled), 17.3–193.9 μg/100 g (fried), and 7.3–160.1 μg/100 g (grilled). Results found significant effects of species and cooking method on Se content (*p* < 0.05). Fried seafood exhibited the highest estimated marginal mean Se concentration (a 78.8 μg/100 g edible portion), significantly higher than other methods. True retention (%TR) of Se ranged from 40.4% to 100%, depending on species and method. Bigfin reef squid, wedge shell, and silver pomfret showed the highest %TR (100%), while splendid squid exhibited the lowest (52.5%). Significant interaction effects on %TR were also observed (*p* < 0.05). Fried seafood had the highest mean %TR (88.8%), followed by grilled (82.1%) and boiled (79.7%). These findings highlight the effects of both species and cooking method on Se retention, emphasizing the nutritional value of selected seafood in preserving bioavailable Se after cooking.

## 1. Introduction

Selenium (Se) is a vital trace element involved in various human body functions [1,2]. Inadequate (Se) intake can lead to abnormal physiological functions and negatively impact quality of life. Long-term Se deficiency in humans has been associated with immune system impairment, the development of cardiovascular diseases, malignant cell proliferation, and growth retardation in children [3,4,5]. On the other hand, previous studies have indicated that adequate Se intake contributes to health promotion by reducing the risk of developing diabetes mellitus, lowering cancer risk through its antioxidant activity, and supporting the immune system by promoting T-cell differentiation [6,7,8].

Assessment of Se status commonly utilizes serum Se concentration, with levels below 63 μg/L considered indicative of deficiency [9]. Clinical manifestations associated with Se deficiency include alopecia, brittle nails, a characteristic garlic-like odor in breath, reproductive dysfunction, and persistent fatigue, among others [10]. Evidence from multiple studies highlights the prevalence of Se deficiency across diverse populations. For instance, a cross-sectional study conducted in Saudi Arabia reported that 41% of adult participants exhibited low Se concentrations in toenail samples (<0.56 μg/g) [11]. In Zambia, research indicated a median plasma Se level of 0.27 μmol/L among adults, which falls below the recommended threshold [12]. Similarly, a study conducted in Spain found that 13.9% of children had serum Se concentrations below 60 μg/L [13]. In the Thai context, it was reported that 56% of children living with HIV had inadequate serum Se levels [14]. Based on the Dietary Reference Intakes (DRIs), the recommended daily intake (RDI) of Se for adults is 55 μg/day [15]. Therefore, estimating selenium intake from seafood should be expressed in relation to Se content per gram of edible portion (e.g., meat or filet) to assess its contribution to daily requirements accurately.

Seafood is widely recognized as a significant dietary source of Se, given its naturally high Se content. A study conducted in Japan reported Se concentrations in alfonsino muscle tissue at 1.27 mg/kg, with levels in internal organs ranging from 1.10 to 24.8 mg/kg. Similarly, Se levels in mullet and Pacific herring were found to range from 1.07 to 1.20 mg/kg [16]. A study in Mexico identified Se concentrations in commercially available shrimp ranging from 0.02 to 3.8 mg/kg [17]. In addition, a study in India also investigated the impact of cooking methods on the nutritional profile, microelements, and Se-Pb risk–benefit of farmed shrimp [18]. The findings gained from such studies are essential for guiding consumers and public health authorities toward safer and more nutritious seafood preparation practices, particularly in regions with high seafood consumption. Thailand, as a global leader in both marine capture fisheries and aquaculture, offers widespread access to a diverse array of seafood species—including shrimp, crabs, and shellfish—at relatively low cost. These seafood items are commonly integrated into traditional Thai diets through a variety of cooking techniques. Despite the nutritional significance of Se and Thailand’s strong seafood industry, the data on Se concentrations in locally available seafood remain scarce, primarily due to analytical challenges such as matrix interferences and the trace-level nature of selenium analysis. To date, only one study in Thailand has reported Se concentrations in commonly consumed freshwater and marine fish, utilizing ICP-MS/MS [19]. Given the critical role of selenium in human health, the widespread consumption of seafood in Thailand, and the limited existing data on its Se content—especially following different culinary preparations—further investigation is warranted. Therefore, the present study aims to enhance the current understanding and establish a more comprehensive database regarding the impact of various cooking methods on the selenium content of seafood commonly consumed in Thailand.

## 2. Materials and Methods

### 2.1. Chemical and Reagents Used in This Study

Analytical-grade reagents were used throughout this study to ensure the accuracy and reliability of results. Suprapur nitric acid (65% HNO_3_) was obtained from Sigma-Aldrich (St. Louis, MO, USA) and used for sample digestion due to its high purity and compatibility with trace element analysis. Standard reference materials for selenium (Se SRM3149) and rhodium (Rh SRM3144) were acquired from the National Institute of Standards and Technology (NIST), Gaithersburg, MD, USA, and employed for the preparation of calibration standards and as an internal standard, respectively. The certified reference material (CRM) SRM1566b (oyster tissue) from NIST and CRM NMIJ7402-a (codfish tissue) from the National Metrology Institute of Japan (NMIJ) were utilized to validate the accuracy and precision of the analytical method. Milli-Q^®^ ultrapure water (Millipore Sigma, MA, USA) was used throughout all study procedures for sample dilution and reagent preparation to avoid contamination and ensure analytical consistency.

### 2.2. Seafood Sample Collection and Preparation

Twenty seafood species, commonly produced and widely available in local markets across Thailand, were selected according to national aquacultural data sources [20,21] and national data on Thai food composition [22]. Table 1 presents the scientific names, common names, and corresponding Thai names of each species. Specimens were randomly procured from 3 to 4 vendors at three local seafood markets in Chonburi (500 g for each purchased sample), a coastal province situated in eastern Thailand. All seafood samples were subjected to preparation and cooking methods in accordance with previously established procedures [23] at the Department of Nutrition and Dietetics, Burapha University.

In brief, each sample was weighed (g) both before and after the removal of inedible components. Deionized water (Milli-Q^®^ EQ 7000, Merck KGaA, Darmstadt, Germany) was used for boiling, palm oil was used for frying, and grilling was performed using an electric grill pan. Cooked samples were homogenized using a homogenizer (Ultra-Turrax T25, IKA, Staufen, Germany), then freeze-dried (Kinetic Engineering Co., LTD., Bangkok, Thailand) under the conditions of −55 °C and 0.040 mbar, for 24 h. After freeze-drying, samples were ground into a fine powder using a laboratory grinder (A11 Basic Analytical Mill, IKA, Staufen, Germany), transferred into screw-capped plastic containers, and stored at −40 °C until further analysis. Moisture content was determined based on the difference in mass before and after freeze-drying.

### 2.3. Total Se Concentration Analysis

The total Se concentration in seafood samples was quantified using ICP-MS/MS (Agilent 8800, Agilent Technologies, Santa Clara, CA, USA) at the National Institute of Metrology, Thailand. To reduce potential spectral interferences and improve both sensitivity and precision in the detection of Se isotopes, analyses were conducted using both collision and reaction gas modes. Calibration curves were established using serial dilutions of Se stock solutions in 2% HNO_3_ to generate external standards. To correct for instrumental variability, a rhodium (Rh) stock solution was prepared and incorporated into each final sample as an internal standard.

Sample preparation followed a modified protocol based on a previously established method [24]. Briefly, 0.5 g of each freeze-dried sample was accurately weighed into glass vials, followed by the addition of 5 mL of Suprapure 65% HNO_3_. The mixture—including blanks, CRMs, and seafood samples—was digested in triplicate using a microwave digestion system (Anton Paar Multiwave 7000, Anton Paar, Graz, Austria) under previously reported conditions [19]. Post-digestion, the solutions were quantitatively transferred to polypropylene tubes and diluted with deionized water to a final volume of 40 mL. An aliquot from each digested sample was then transferred into a new polypropylene tube, spiked with the Rh internal standard, and adjusted to a final volume of 10 mL. The prepared samples were subsequently analyzed for Se content using ICP-MS/MS.

### 2.4. Analytical Method Accuracy and Validation

To ensure the reliability of Se quantification via ICP-MS/MS, CRMs—SRM 1566b (oyster tissue) and NMIJ 7402-a (codfish tissue)—were subjected to triplicate digestion and analyzed using the same instrumental conditions as the test samples. The measured Se concentrations were adjusted to a dry mass basis, accounting for sample moisture content, in accordance with the certified values. To evaluate the analytical accuracy, the experimentally obtained Se concentrations were statistically compared with the corresponding certified values.

In this study, the Se concentration determined for SRM 1566b was 2.11 ± 0.10 mg/kg, compared with its certified value of 2.06 ± 0.15 mg/kg. For NMIJ 7402-a, the measured concentration was 1.78 ± 0.05 mg/kg, relative to the certified value of 1.80 ± 0.20 mg/kg. The precision of the measurements, expressed as relative standard deviation (RSD), was 4.98% for SRM 1566b and 2.69% for NMIJ 7402-a. These results confirm that the employed analytical method exhibited acceptable accuracy and precision for the determination of Se in biological matrices.

For the limit of detection (LOD) and limit of quantitation (LOQ), the LOD refers to the lowest concentration of Se that can be reliably detected in the seafood samples analyzed in this study. The LOD was determined using the following equation: (1)$$LOD\;=\;\frac{3\;\times \;standard\;deviation}{slope}$$

The LOQ refers to the lowest amount of Se that can be quantitatively determined in the seafood samples with acceptable precision and accuracy. The LOQ was determined using the following equation:
(2)$$LOQ\;=\;\frac{10\;\times \;standard\;deviation}{slope}$$

Validation of the concentration corresponding to the LOQ is essential to ensure that measurements at this level exhibit acceptable accuracy and precision. To verify the LOQ, a Se standard was spiked into blank solutions (*n* = 10), and the samples were analyzed using ICP-MS/MS. In the present study, the LOQ was determined to be 3 μg/kg, with a recovery rate of 104.2% and an RSD of 7.3%. These values comply with established criteria for LOQ validation as outlined in previous guidelines [25,26]. The validated LOQ was subsequently applied to the analysis of Se concentrations in the seafood samples using the following equation:
(3)$$\mathrm{LOQ}\;\mathrm{for}\;\mathrm{solid}\;\mathrm{sample}\;=\;LOQ\;\times \;\frac{mass\;of\;digested\;sample\;solution}{mass\;of\;sample}$$

### 2.5. Percentage of Yield Factor Determination

The mass yield factor (YF) quantifies the changes in each seafood sample mass resulting from cooking, specifically accounting for the loss or gain of water and/or fat. It was calculated by dividing the mass of each cooked seafood sample (g) by the mass of the raw sample (g).

### 2.6. True Retention of Se in Cooked Seafood

To assess the effect of each cooking method on selenium (Se) retention in the seafood samples, all specimens were accurately weighed to three significant digits using an analytical balance, with measurements recorded both before and after the cooking process. The true retention (TR) of Se in the cooked samples was calculated using the following equation:
(4)$$\%\mathrm{TR}\;=\;\frac{\mu g\;Seper\;100\;g\;of\;cooked\;fish\;\times \;mass\;of\;cooked\;fish}{\mu g\;Seper\;100\;g\;of\;raw\;fish\times mass\;of\;raw\;fish}\times 100$$

### 2.7. Statistical Analyses

Data regarding the percentages of edible portions, yield factors, moisture content, and Se concentrations in raw and cooked (boiled, fried, and grilled) seafood samples were expressed as mean ± standard deviation (SD). Variations in Se concentrations among different seafood types and their corresponding true retention following cooking were analyzed using two-way analysis of variance (ANOVA), including interaction effects. Post hoc multiple comparisons were performed using Tukey’s Honestly Significant Difference (HSD) test. All statistical analyses were conducted using IBM^®^ SPSS Statistics for Windows, Version 24.0, with a *p*-value of <0.05 considered indicative of statistical significance.

## 3. Results

### 3.1. Edible Portions, Yield Factors, and Moisture Content

The edible portion (EP), yield factor, and moisture content were not subjected to statistical analysis, as they serve as fixed reference parameters rather than experimental variables. These values are obtained from established sources and applied consistently to facilitate accurate nutrient conversion and standardized data processing. Their primary function is to ensure consistency and comparability in nutrient estimation, rather than to be used for hypothesis testing.

In this study, the EP of each seafood species was determined based on wet mass after the removal of inedible components such as shells, scales, bones, and peels. The results revealed a wide range of EP values among the selected seafood species, varying from 10.0 to 92.7%. When analyzed by group, the EP ranged from 42.1% to 78.9% for shrimp and lobster in their fresh form, 10.0–31.4% for crabs, 84.1–92.7% for squids, 16.6–88.3% for mollusks, 39.7% for horseshoe crab, and 39.5–54.0% for marine fish (Table 2). The yield factor of food refers to the ratio of the EP to the original mass (as purchased) of a food item. It reflects the amount of usable food remaining after the removal of inedible or waste components during preparation. The results showed that the yield factors of seafood varied by cooking method: boiling yielded values between 0.5 and 0.9, frying between 0.3 and 0.8, and grilling between 0.2 and 0.8. Moisture content was also determined as a critical parameter in this study, as it directly influences both the mass-based calculation of Se concentration and the interpretation of %TR values. The moisture level of seafood can vary significantly with cooking methods, affecting nutrient density on a per-mass basis. By assessing moisture content, the data could be standardized to a consistent basis (e.g., per 100 g), thereby enabling more accurate comparisons of Se concentrations across raw and cooked samples. The analysis showed that moisture content was lowest in fried samples (38.8–78.0 g/100 g), followed by grilled (49.5–85.7 g/100 g), boiled (60.6–84.2 g/100 g), and fresh samples (64.1–88.0 g/100 g), as presented in Table 2.

### 3.2. Selenium Concentration in Seafood Samples

The total Se concentrations in seafood species prepared using different cooking methods are presented in Table 2. The Indo-Pacific horseshoe crab exhibited the highest Se concentrations across all cooking methods, with values of 155.0 μg/100 g in the fresh product, 106.6 μg/100 g in the boiled product, 193.9 μg/100 g in the fried product, and 160.1 μg/100 g in the grilled product. The Se concentration in fresh seafood ranged from 8.6 to 155.5 μg/100 g, in boiled seafood from 14.3 to 106.6 μg/100 g, in fried seafood from 17.3 to 193.9 μg/100 g, and in grilled seafood from 7.3 to 160.1 μg/100 g (Table 2).

A two-way ANOVA with interaction, followed by Tukey’s HSD post hoc test, revealed a statistically significant interaction between seafood species and cooking methods on Se content (*p* < 0.05; Figure 1A). Notably, Se levels varied significantly among the different seafood species (*p* < 0.05; Table 3 and Table 4), with the Indo-Pacific horseshoe crab exhibiting the highest Se concentration (estimated marginal mean: 154.1 μg per 100 g of EP), significantly exceeding those of other seafood species (range: 12.1–96.2 μg per 100 g of EP). Furthermore, the cooking method had a significant impact on Se content (*p* < 0.001; Table 4), with fried seafood showing the highest Se levels (estimated marginal mean: 78.8 μg per 100 g of EP), significantly greater than those observed in fresh (50.8 μg), boiled (54.8 μg), and grilled (71.6 μg) samples.

### 3.3. Effects of Different Cooking Methods on TR of Se

The TR data of Se are presented in Table 3 and Table 4. The TR of boiled seafood ranged from 47.4% to 100%, fried seafood ranged from 40.4% to 100%, and grilled seafood ranged from 58.6% to 100%. Bigfin reef squid, wedge shell, and silver pomfret exhibited the highest Se true retention (100%TR) across all cooking methods. In contrast, splendid squid showed the lowest average TR, with a value of 52.5% (Table 3 and Table 4).

Two-way ANOVA with interaction effects, followed by Tukey’s HSD post hoc analysis, revealed statistically significant differences in the percentage of %TR as influenced by the interaction between seafood species and cooking methods. As illustrated in Figure 1B, the impact of cooking methods on %TR varies across different seafood species (*p* < 0.001). For example, boiled Serrated Mud Crab (36.6%TR) and boiled ornate rock lobster (47.4%TR) demonstrated lower %TR values compared to other boiled seafood (Table 2). Additionally, true retention for other boiled species such as Musk Crab (65.7%TR), blue crab (61.5%TR), and splendid squid (54.5%TR) was lower than that observed in red frog crab, bigfin reef squid, northern whiting fish, and silver pomfret. Among fried seafood, splendid squid exhibited the lowest %TR (40.4%TR), while most other fried samples showed values exceeding 70%. For grilled seafood, splendid squid (62.3%TR) and oysters (58.6%TR) had lower retention compared to other grilled species. Furthermore, Giant Tiger Prawn (67.3%TR), ornate rock lobster (62.7%TR), red frog crab (63.4%TR), and cockle (66.2%TR) had lower %TR than banana prawn, bigfin reef squid, wedge shell, and silver pomfret (Table 2 and Table 3).

A significant effect was observed for both seafood species and cooking methods on the %TR (*p* < 0.05). Banana prawn, bigfin reef squid, razor clam, mussels, wedge shell, northern whiting fish, and silver pomfret exhibited significantly higher %TR values (*p* < 0.05), with estimated marginal means ranging from 91.8% to 100%, compared to other seafood species (52.5–87.8%) (Table 4). Regarding cooking methods, significant differences in %TR were also found among boiling, frying, and grilling. Fried seafood demonstrated the highest %TR (*p* < 0.05), with an estimated marginal mean of 88.8%, compared to boiled (79.7%) and grilled (82.1%) seafood (Table 4).

## 4. Discussion

### 4.1. Edible Portions, Yield Factors, and Moisture Content

The EP of seafood in this study varied depending on the species. Squids exhibited a higher EP compared to shellfish, which can be attributed to the absence of shells or peels in squids; only a few internal organs need to be removed. In contrast, shellfish such as crabs and shrimps possess a substantial amount of inedible shell, resulting in a greater proportion being discarded during preparation, particularly following traditional Thai cooking methods. In addition, the EP values of seafood observed in this study were consistent with previously reported data. For example, EP values typically range from 80 to 93% for squids, 45 to 60% for shrimps/prawns, 10 to 30% for crabs, 20 to 30% for lobsters, 20 to 35% for mussels/clams/oysters, and 35 to 45% for horseshoe crab [27,28]. For fish, northern whiting and silver pomfret prepared according to traditional Thai cooking methods exhibited lower EP values compared to local fish in Bangladesh, which have been reported to range from 62% to 85% [29]. The variability in the yield factor may be influenced by differences in the tissue composition and moisture content of the seafood, both of which significantly affect mass changes during cooking [30]. Additionally, prior reports have highlighted that variations in fat content can also contribute to differences in the yield factor [31]. The findings related to moisture content were consistent with the observed high moisture loss in fried and grilled seafood compared to boiled samples. This can be attributed to the high-temperature processing involved in frying and grilling, which is associated with moisture evaporation—particularly in seafood, where water constitutes the majority of the raw mass [32,33,34]. In this study, a negative correlation was observed between cooking temperature and moisture content, indicating that higher cooking temperatures significantly contributed to greater moisture loss across all seafood types analyzed.

### 4.2. Selenium Concentration

This study confirmed that seafood commonly consumed in Thailand contains high concentrations of Se, with levels comparable to those reported in previous studies on marine fish. For example, research conducted in Europe documented Se concentrations in fish ranging from 22 to 61 μg/100 g [35], while studies from Japan reported a broader range between 12 and 127 μg/100 g [16]. These findings reinforce existing evidence that seafood is a valuable dietary source of Se and should be promoted for inclusion in the diets of both the general population and individuals at risk of Se deficiency.

Additionally, this study demonstrated that Se concentrations in seafood are significantly influenced by cooking methods. Among the methods evaluated, fried seafood exhibited the highest Se levels, followed by grilled, boiled, and fresh preparations. This trend may be attributed to the thermal stability of Se at high temperatures encountered during frying and grilling [36], whereas boiling likely facilitates Se loss through leaching into the cooking water. The chemical forms of Se present in seafood may also influence its stability during cooking. Predominantly, Se exists in the forms of selenomethionine and selenocysteine in fish and other seafood [37], although some species, such as tuna, contain selenoneine as the primary Se compound in their blood [38]. These low-molecular-mass Se species are water-soluble and may be readily lost during cooking, especially via boiling [39]. This could explain why boiling leads to a greater reduction in Se content compared to frying. Furthermore, the higher Se concentrations observed in fried samples may also be associated with a substantial loss of moisture during high-temperature cooking, leading to a relative concentration of remaining nutrients, including Se. As moisture decreases, the centesimal composition of the food shifts, resulting in increased proportions of dry matter such as proteins, lipids, and micronutrients per unit mass [32,33]. In addition, the absorption of frying oil—potentially containing trace amounts of Se—may contribute to elevated Se levels in cooked samples. Although the composition of the oil was not analyzed in this study, this variable should be considered in future research to clarify its potential role in Se enrichment.

Statistical analysis further revealed significant interaction effects between seafood species and cooking methods on Se concentrations. Specifically, the Indo-Pacific horseshoe crab exhibited significantly higher Se levels than other seafood species. These findings are consistent with a previous study showing that the cooking method influences Se content in both freshwater and marine fish, with frying generally yielding the highest concentrations [19]. Interestingly, this effect contrasts with findings for other nutrients; for instance, a study on Thai fish reported no significant impact of cooking method on vitamin D content [23]. The observed differences in Se loss among seafood species may be attributed to several intrinsic biological and structural factors, such as fat content, protein structure, and water-holding capacity, which can influence the degree to which Se is retained or lost during thermal processing [32,33].

Notably, species with firmer muscle structures and higher lipid content may exhibit greater Se retention, particularly during high-temperature cooking methods such as frying or grilling. This may be due to the physical entrapment of Se within hydrophobic compartments or associations with heat-stable proteins, which reduce Se leaching into cooking media [40]. Moreover, the impact of temperature is closely linked to the physical state of the seafood matrix. Higher temperatures typically accelerate moisture loss, which leads to a concentration of remaining nutrients, including Se. However, prolonged or excessive heat can also disrupt Se–protein complexes or promote oxidation, potentially reducing Se bioavailability or causing volatilization of certain Se species [41]. For example, selenomethionine and selenocysteine—major organic Se forms—can degrade under intense heat, although to varying extents depending on matrix and cooking duration [42]. Thus, the observed interactions between seafood type and cooking method likely reflect a combination of species-specific composition and thermal stability of Se compounds. Understanding these interactions is essential not only for accurate Se quantification but also for maximizing the nutritional benefits of seafood in typical Thai cooking practices.

### 4.3. Influence of Various Cooking Methods on the True Retention of Selenium

The majority of seafood samples analyzed exhibited a %TR of Se exceeding 60%, with the exception of splendid squid, which demonstrated a %TR of 52.5%. These results suggest that Se is generally well-retained across various seafood types subjected to different cooking methods. Nonetheless, variations in the physical properties and tissue composition of the seafood may influence Se retention following thermal processing. For instance, fish species possessing scales may experience reduced heat penetration during cooking, potentially enhancing Se preservation [43,44]. Additionally, previous studies have indicated that the high cholesterol content in squid may interact with Se in a manner that affects its retention [45], which may account for the relatively low %TR observed in splendid squid. The %TR values observed in this study are consistent with those reported for both freshwater and marine fish in Thailand, which range from 60% to 100% [19]. The findings suggest that commonly consumed seafood in Thailand maintains a high Se retention rate across a variety of cooking methods. Moreover, the results align with data published by the Food and Agriculture Organization of the United Nations (FAO), which reported Se retention in marine fish and seafood within the 90–100% range [46], highlighting the element’s relative stability under heat exposure. Statistical analysis revealed significant differences in the interaction effects between seafood species and cooking methods on Se retention. These findings underscore that different cooking methods have varying impacts on the %TR of Se, depending on the specific characteristics of each seafood species.

Frying and grilling were associated with higher percentages of Se true retention (%TR) compared to boiling, suggesting that Se demonstrates considerable heat stability. The lower %TR observed in boiling may be attributed to Se losses through leaching into the cooking water. Nevertheless, the %TR of Se in boiled seafood observed in this study is comparable to findings from similar water-based cooking methods in Europe. For example, steamed Gilthead seabream (*Sparus aurata*) exhibited Se %TR values ranging from 90% to 100% [47]. These findings indicate that boiling, despite potential leaching, can still result in high Se retention in seafood commonly consumed in Thailand.

Although the results revealed a good source of Se from seafood in this study, and seafood consumption should be promoted, previous reports have identified the occurrence of heavy metal contamination in a range of fish species and other seafood [48], with mercury (Hg) reported as the predominant contaminant. Hg frequently forms complexes with Se, and the Se: Hg molar ratio in fish has been documented to range between 0.23 and 1 [49,50]. In addition to Hg, other heavy metals such as cadmium (Cd), arsenic (As), and lead (Pb) are well-established toxicants known to pose significant health risks [51], including an elevated likelihood of cancer [52]. Such studies are essential to ensure that contamination levels remain within the maximum allowable limits established by the Food and Agriculture Organization (FAO), including Pb < 0.3 mg/kg and Hg within the range of 1.2–1.6 mg/kg [53]. Although earlier studies have shown that fish commonly consumed in Thailand contain low levels of these toxic heavy metals [54], the specific seafood species examined in the present study have not yet been assessed. Therefore, further investigation is warranted to determine the concentrations of these heavy metals in commonly consumed seafood in Thailand. A notable limitation of the present study is the lack of Se speciation and Se bioaccessibility analysis. Accordingly, future investigations should focus on identifying and quantifying the different chemical forms of Se and assessing their bioaccessibility in the selected seafood species.

## 5. Conclusions

This study demonstrates that both seafood species and cooking methods significantly affect Se content and %TR. The Indo-Pacific horseshoe crab showed the highest Se concentration, while frying consistently resulted in greater Se levels and retention compared to boiling and grilling. Certain species, such as banana prawn, bigfin reef squid, and silver pomfret, exhibited high Se retention across all cooking methods, indicating their nutritional benefit. These findings support the role of seafood and cooking practices in enhancing dietary Se intake and can inform public health recommendations.

## Figures and Tables

**Figure 1 foods-14-02700-f001:**
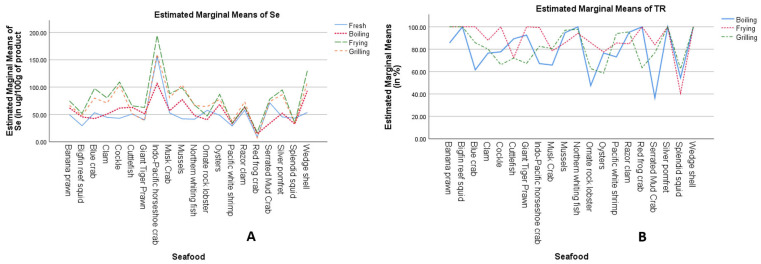
Combined influence of seafood species and culinary techniques on Se concentration (**A**) and true retention percentage (**B**).

**Table 1 foods-14-02700-t001:** The selected 20 commonly consumed seafood types used in this study.

Common Name	Scientific Name	Local Name	Purchase
			(Month/Year)
Shrimp and prawn (captured)			
Pacific white shrimp	*Litopenaeus vannamei*	Koong Khaw	2/2025
Banana prawn	*Fenneropenaeus merguiensis*	Koong Share Buay	2/2025
Giant Tiger Prawn	*Penaeus monodon*	Koong Kula Dam	2/2025
Ornate rock lobster	*Panulirus ornatus*	Koong Mungkorn	2/2025
Crabs (captured)			
Musk Crab	*Charybdis feriata Linnaeus*	Pu Lai Sua	2/2025
Blue crab	*Portunus pelagicus*	Pu Ma	3/2025
Serrated Mud Crab	*Scylla serrata*	Pu Dam	3/2025
Red frog crab	*Ranina vanima*	Pu Juckajun	3/2025
Squids (captured)			
Splendid squid	*Loligo duvauceli*	Pla Muek Kluay	3/2025
Cuttlefish	*Sepia brevimana*	Pla Muek Kradong	3/2025
Bigfin reef squid	*Sepioteuthis lessoniana*	Pla Muek Hom	3/2025
Shellfish (captured)			
Razor clam	*Solen strictus Gould*	Hoi Lord	3/2025
Oysters	*Crassostrea gigas*	Hoi Nang Rom	3/2025
Cockle	*Tegillarca granosa*	Hoi Krang	3/2025
Clam	*Paphia undulata*	Hoi Lai	3/2025
Mussels	*Perna viridis*	Hoi Malang Poo	3/2025
Wedge shell	*Mercenaria mercenaria*	Hoi Talab	3/2025
Indo-Pacific horseshoe crab (eggs)	*Tachypleus gigas*	Mangda Jan	3/2025
Marine fish (farmed)			
Northern whiting fish	*Sillago sihama*	Pla Hed Kone	3/2025
Silver pomfret	*Pampus argenteus*	Pla Jaramed Khaw	3/2025

**Table 2 foods-14-02700-t002:** The percentages of edible portion, yield factor, and moisture content.

Seafood Name	Type of Cooked Sample	Edible Portion (%)	Yield Factor	Moisture (g/100 g)	Se Concentration (μg/100 g of Product)	TR of Se (%) ^
Pacific white shrimp	Fresh	60.2 ± 0.1	-	76.1 ± 0.0	28.9 ± 0.2	-
	Boiled	60.4 ± 0.3	0.6 ± 0.0	70.4 ± 0.0	32.9 ± 0.2	73.0 ± 0.6
	Fried	50.4 ± 0.2	0.7 ± 0.0	65.1 ± 0.3	34.0 ± 0.7	85.5 ± 0.9
	Grilled	52.3 ± 0.2	0.7 ± 0.0	67.6 ± 0.7	37.7 ± 0.2	94.0 ± 0.3
Banana prawn	Fresh	78.9 ± 0.9	-	81.5 ± 0.9	49.5 ± 0.3	-
	Boiled	54.9 ± 0.5	0.6 ± 0.0	71.8 ± 0.1	61.9 ± 0.4	85.6 ± 0.0
	Fried	34.5 ± 0.4	0.7 ± 0.0	68.8 ± 0.0	74.7 ± 0.8	100.0 ± 0.0
	Grilled	54.9 ± 0.5	0.7 ± 0.0	74.6 ± 0.0	66.5 ± 0.8	100.0 ± 0.0
Giant Tiger Prawn	Fresh	57.0 ± 0.2	-	84.4 ± 0.6	40.1 ± 0.1	-
	Boiled	59.2 ± 0.1	0.7 ± 0.0	74.1 ± 0.9	51.4 ± 0.2	92.5 ± 0.7
	Fried	52.0 ± 0.3	0.6 ± 0.0	69.4 ± 0.4	62.8 ± 0.8	100.0 ± 0.0
	Grilled	55.1 ± 0.1	0.6 ± 0.0	75.6 ± 0.7	38.7 ± 0.4	67.3 ± 0.5
Ornate rock lobster	Fresh	42.1 ± 0.6	-	79.4 ± 0.3	57.5 ± 0.9	-
	Boiled	38.1 ± 0.6	0.6 ± 0.0	79.8 ± 0.0	40.4 ± 0.0	47.4 ± 0.7
	Fried	35.8 ± 0.5	0.8 ± 0.0	78.0 ± 0.4	47.1 ± 0.5	85.7 ± 0.8
	Grilled	38.8 ± 0.6	0.5 ± 0.0	68.3 ± 0.3	64.3 ± 0.8	62.7 ± 0.1
Musk Crab	Fresh	30.2 ± 0.7	-	82.7 ± 0.1	52.7 ± 0.1	-
	Boiled	33.3 ± 0.6	0.6 ± 0.0	83.3 ± 0.3	56.9 ± 0.0	65.7 ± 0.9
	Fried	31.7 ± 0.6	0.4 ± 0.0	58.3 ± 0.3	87.9 ± 0.9	78.5 ± 0.5
	Grilled	33.4 ± 0.7	0.5 ± 0.0	75.9 ± 0.2	80.9 ± 0.1	80.4 ± 0.7
Blue crab	Fresh	19.1 ± 0.8	-	81.3 ± 0.5	53.1 ± 0.9	-
	Boiled	24.0 ± 0.4	0.7 ± 0.0	83.3 ± 0.3	42.3 ± 0.5	61.5 ± 0.7
	Fried	18.2 ± 0.8	0.5 ± 0.0	48.6 ± 0.4	97.4 ± 0.5	100.0 ± 0.0
	Grilled	19.6 ± 0.6	0.5 ± 0.0	66.6 ± 0.6	79.8 ± 0.9	85.8 ± 0.3
Serrated Mud Crab	Fresh	31.4 ± 0.7	-	75.2 ± 0.3	71.2 ± 0.8	-
	Boiled	33.6 ± 0.7	0.7 ± 0.0	84.2 ± 0.1	33.6 ± 0.8	36.6 ± 0.2
	Fried	30.3 ± 0.6	0.7 ± 0.0	61.3 ± 0.6	78.0 ± 0.2	83.9 ± 0.1
	Grilled	32.4 ± 0.6	0.7 ± 0.0	72.5 ± 0.0	73.8 ± 0.0	76.7 ± 0.5
Red frog crab	Fresh	10.0 ± 0.0	-	83.3 ± 0.3	8.6 ± 0.8	-
	Boiled	11.1 ± 0.1	0.7 ± 0.0	72.7 ± 0.2	14.9 ± 0.4	100.0 ± 0.0
	Fried	10.6 ± 0.0	0.6 ± 0.0	58.3 ± 0.3	17.3 ± 0.6	100.0 ± 0.0
	Grilled	10.9 ± 0.0	0.7 ± 0.0	85.7 ± 0.1	7.3 ± 0.5	63.4 ± 0.6
Splendid squid	Fresh	84.7 ± 0.3	-	82.7 ± 0.9	43.1 ± 0.6	-
	Boiled	87.4 ± 0.5	0.7 ± 0.0	83.6 ± 0.7	32.4 ± 0.6	54.5 ± 0.3
	Fried	85.7 ± 0.4	0.5 ± 0.0	72.2 ± 0.2	34.4 ± 0.0	40.4 ± 0.6
	Grilled	85.8 ± 0.4	0.8 ± 0.0	82.1 ± 0.4	33.6 ± 0.5	62.3 ± 0.8
Cuttlefish	Fresh	84.1 ± 0.0	-	83.5 ± 0.2	50.8 ± 0.0	-
	Boiled	89.1 ± 0.2	0.7 ± 0.0	79.6 ± 0.6	63.0 ± 0.9	89.3 ± 0.0
	Fried	82.0 ± 0.0	0.5 ± 0.0	73.0 ± 0.7	65.8 ± 0.3	72.4 ± 0.5
	Grilled	80.7 ± 0.1	0.7 ± 0.0	79.3 ± 0.6	51.5 ± 0.8	72.0 ± 0.5
Bigfin reef squid	Fresh	92.7 ± 0.2	-	87.2 ± 0.7	29.4 ± 0.3	-
	Boiled	95.2 ± 0.1	0.7 ± 0.0	81.5 ± 0.7	45.5 ± 0.9	100.0 ± 0.0
	Fried	91.3 ± 0.3	0.6 ± 0.0	77.9 ± 0.4	51.7 ± 0.9	100.0 ± 0.0
	Grilled	92.9 ± 0.2	0.6 ± 0.0	80.0 ± 0.0	48.4 ± 0.6	100.0 ± 0.0
Razor clam	Fresh	88.3 ± 0.9	-	78.6 ± 0.2	57.8 ± 0.0	-
	Boiled	89.1 ± 0.9	0.8 ± 0.0	73.9 ± 0.8	63.4 ± 0.5	95.2 ± 0.3
	Fried	84.6 ± 0.8	0.7 ± 0.0	67.0 ± 0.0	64.5 ± 0.4	84.7 ± 0.9
	Grilled	85.5 ± 0.9	0.7 ± 0.0	71.0 ± 0.0	73.4 ± 0.0	95.4 ± 0.7
Oysters	Fresh	68.2 ± 0.8	-	78.6 ± 0.7	48.6 ± 0.0	-
	Boiled	71.3 ± 0.7	0.5 ± 0.0	73.2 ± 0.1	68.4 ± 0.0	76.6 ± 0.3
	Fried	66.2 ± 0.6	0.4 ± 0.0	58.9 ± 0.4	86.7 ± 0.7	77.5 ± 0.4
	Grilled	67.0 ± 0.4	0.3 ± 0.0	66.2 ± 0.5	77.5 ± 0.0	58.6 ± 0.3
Cockle	Fresh	16.6 ± 0.8	-	88.0 ± 0.0	42.9 ± 0.5	-
	Boiled	17.5 ± 0.7	0.5 ± 0.0	80.0 ± 0.0	61.7 ± 0.1	77.7 ± 0.3
	Fried	15.4 ± 0.2	0.4 ± 0.0	66.6 ± 0.6	109.2 ± 0.5	100.0 ± 0.0
	Grilled	16.7 ± 0.7	0.2 ± 0.0	70.5 ± 0.8	103.7 ± 0.2	66.2 ± 0.1
Clam	Fresh	28.1 ± 0.8	-	81.7 ± 0.2	44.6 ± 0.8	-
	Boiled	30.2 ± 0.7	0.6 ± 0.0	76.4 ± 0.7	50.5 ± 0.9	76.5 ± 0.3
	Fried	24.6 ± 0.5	0.4 ± 0.0	58.6 ± 0.9	80.2 ± 0.2	88.0 ± 0.1
	Grilled	27.6 ± 0.8	0.5 ± 0.0	68.7 ± 0.5	70.8 ± 0.7	80.0 ± 0.7
Mussels	Fresh	31.1 ± 0.5	-	86.5 ± 0.1	42.0 ± 0.3	-
	Boiled	37.9 ± 0.6	0.5 ± 0.0	78.5 ± 0.7	77.1 ± 0.6	94.8 ± 0.4
	Fried	33.3 ± 0.3	0.3 ± 0.0	69.0 ± 0.4	98.8 ± 0.3	85.6 ± 0.7
	Grilled	33.2 ± 0.3	0.3 ± 0.0	72.7 ± 0.2	104.0 ± 0.2	97.0 ± 0.0
Wedge shell	Fresh	29.3 ± 0.6	-	77.0 ± 0.4	53.5 ± 0.2	-
	Boiled	29.0 ± 0.6	0.6 ± 0.0	65.6 ± 0.2	93.2 ± 0.5	100.0 ± 0.0
	Fried	27.4 ± 0.5	0.5 ± 0.0	53.3 ± 0.3	129.7 ± 0.0	100.0 ± 0.0
	Grilled	28.3 ± 0.5	0.6 ± 0.0	55.5 ± 0.5	108.4 ± 0.1	100.0 ± 0.0
Indo-Pacific horseshoe crab	Fresh	39.7 ± 0.**7**	-	64.1 ± 0.2	155.5 ± 0.8	-
	Boiled	47.7 ± 0.9	0.9 ± 0.0	60.6 ± 0.8	106.6 ± 0.7	67.2 ± 0.2
	Fried	38.1 ± 0.7	0.7 ± 0.0	38.9 ± 0.3	193.9 ± 0.5	99.3 ± 0.9
	Grilled	37.5 ± 0.7	0.8 ± 0.0	49.5 ± 0.9	160.1 ± 0.1	82.5 ± 0.8
Northern whiting fish Silver pomfret	Fresh	39.5 ± 0.3	-	75.2 ± 0.8	41.3 ± 0.9	-
	Boiled	44.3 ± 0.4	0.9 ± 0.0	76.1 ± 0.9	47.9 ± 0.3	100.0 ± 0.0
	Fried	38.6 ± 0.3	0.5 ± 0.0	38.8 ± 0.8	67.1 ± 0.7	94.2 ± 0.3
	Grilled	36.7 ± 0.2	0.6 ± 0.0	61.4 ± 0.0	66.3 ± 0.0	97.7 ± 0.9
Silver pomfret	Fresh	54.0 ± 0.9	-	80.0 ± 0.0	45.3 ± 0.8	-
	Boiled	59.0 ± 0.8	0.9 ± 0.0	75.5 ± 0.1	52.3 ± 0.7	100.0 ± 0.0
	Fried	52.3 ± 0.6	0.5 ± 0.0	52.3 ± 0.8	94.9 ± 0.3	100.0 ± 0.0
	Grilled	53.8 ± 0.6	0.7 ± 0.0	65.5 ± 0.9	86.1 ± 0.8	100.0 ± 0.0

^ TR values above 100% are shown as calculated. Potential Se contribution from frying oil has not been analyzed in this study. Results were determined for three individual samples of each seafood species, with results reported as mean ± standard deviation (SD) (*n* = 3).

**Table 3 foods-14-02700-t003:** Estimated marginal means derived from the interaction between seafood species and culinary techniques on Se content and true selenium retention.

Common Name	Se Content (μg/100 g of Product, Mean ± Standard Error)			TR of Se (%, Mean ± Standard Error)		
	Boiled	Fried	Grilled	Boiled	Fried	Grilled
Pacific white shrimp	32.9 ± 0.2 ^f,g^	34.0 ± 0.7 ^g,h^	37.7 ± 0.2 ^f,g^	73.0 ± 0.6 ^a,b,c^	85.5 ± 0.9 ^a,b^	94.0 ± 0.3 ^a,b^
Banana prawn	61.9 ± 0.40 ^c,d,e,f^	74.7 ± 0.8 ^d,e,f^	66.5 ± 0.8 ^d,e,f^	85.6 ± 0.0 ^a,b^	100.0 ± 0.0 ^a^	100.0 ± 0.0 ^a^
Giant Tiger Prawn	51.4 ± 0.2 ^d,e,f^	62.8 ± 0.8 ^e,f,g^	38.7 ± 0.4 ^f,g^	92.5 ± 0.7 ^a^	100.0 ± 0.0 ^a^	67.3 ± 0.5 ^b,c^
Ornate rock lobster	40.4 ± 0.0 ^e,f,g^	47.1 ± 0.5 ^g,h^	64.3 ± 0.8 ^d,e,f^	47.4 ± 0.7 ^c,d^	85.7 ± 0.8 ^a,b^	62.7 ± 0.1 ^b,c^
Musk Crab	56.9 ± 0.0 ^c,d,e,f^	87.9 ± 0.9 ^d,e,f^	80.9 ± 0.1 ^b,c,d^	65.7 ± 0.9 ^b,c^	78.5 ± 0.5 ^a,b,c^	80.4 ± 0.7 ^a,b,c^
Blue crab	42.3 ± 0.5 ^e,f,g^	97.4 ± 0.5 ^b,c,d^	79.8 ± 0.9 ^b,c,d^	61.5 ± 0.7 ^b,c^	100.0 ± 0.0 ^a^	85.8 ± 0.3 ^a,b,c^
Serrated Mud Crab	33.6 ± 0.8 ^f,g^	78.0 ± 0.2 ^d,e,f^	73.8 ± 0.0 ^c,d,e^	36.6 ± 0.2 ^d^	83.9 ± 0.1 ^a,b^	76.7 ± 0.5 ^a,b,c^
Red frog crab	14.9 ± 0.4 ^g^	17.3 ± 0.6^h^	7.3 ± 0.5 ^g^	100.0 ± 0.0 ^a^	100.0 ± 0.0 ^a^	63.4 ± 0.6 ^b,c^
Splendid squid	32.4 ± 0.6 ^f,g^	34.4 ± 0.0 ^g,h^	33.6 ± 0.5 ^f,g^	54.5 ± 0.3 ^c,d^	40.4 ± 0.6 ^d^	62.3 ± 0.8 ^b,c^
Cuttlefish	63.0 ± 0.90 ^c,d,e^	65.8 ± 0.3 ^e,f,g^	51.5 ± 0.8 ^e,f,g^	89.3 ± 0.0 ^a,b^	72.4 ± 0.5 ^a,b,c,d^	72.0 ± 0.5 ^a,b,c^
Bigfin reef squid	45.5 ± 0.9 ^e,f,g^	51.7 ± 0.9 ^f,g,h^	48.4 ± 0.6 ^e,f,g^	100.0 ± 0.0 ^a^	100.0 ± 0.0 ^a^	100.0 ± 0.0 ^a^
Razor clam	63.4 ± 0.50 ^c,d,e^	64.5 ± 0.4 ^e,f,g^	73.4 ± 0.0 ^c,d,e^	95.2 ± 0.3 ^a^	84.7 ± 0.9 ^a,b^	95.4 ± 0.7 ^a^
Oysters	68.4 ± 0.0 ^c,d,e^	86.7 ± 0.7 ^d,e,f^	77.5 ± 0.0 ^c,d,e^	76.6 ± 0.3 ^a,b,c^	77.5 ± 0.4 ^a,b,c^	58.6 ± 0.3 ^c^
Cockle	61.7 ± 0.10 ^c,d,e,f^	109.2 ± 0.5 ^b,c,d^	103.7 ± 0.2 ^b,c^	77.7 ± 0.3 ^a,b,c^	100.0 ± 0.0	66.2 ± 0.1 ^b,c^
Clam	50.5 ± 0.9 ^d,e,f^	80.2 ± 0.2 ^d,e,f^	70.8 ± 0.7 ^c,d,e^	76.5 ± 0.3 ^a,b,c^	88.0 ± 0.1 ^a,b^	80.0 ± 0.7 ^a,b,c^
Mussels	77.1 ± 0.6 ^b,c,d^	98.8 ± 0.3 ^b,c,d^	104.0 ± 0.2 ^b,c^	94.8 ± 0.4 ^a^	85.6 ± 0.7 ^a,b^	97.0 ± 0.0 ^a^
Wedge shell	93.2 ± 0.5 ^a,b,c^	129.7 ± 0.0 ^b,c^	108.4 ± 0.1 ^b,c^	100.0 ± 0.0 ^a^	100.0 ± 0.0 ^a^	100.0 ± 0.0 ^a^
Indo-Pacific horseshoe crab	106.6 ± 0.7 ^a,b^	193.9 ± 0.5 ^a^	160.1 ± 0.1 ^a^	67.2 ± 0.2 ^b,c^	99.3 ± 0.9 ^a^	82.5 ± 0.8 ^a,b,c^
Northern whiting fish	47.9 ± 0.3 ^d,e,f,g^	67.1 ± 0.7 ^e,f,g^	66.3 ± 0.0 ^d,e,f^	100.0 ± 0.0 ^a^	94.2 ± 0.3 ^a^	97.7 ± 0.9 ^a^
Silver pomfret	52.3 ± 0.7 ^d,e,f^	94.9 ± 0.3 ^b,c,d^	86.1 ± 0.8 ^b,c,d^	100.0 ± 0.0 ^a^	100.0 ± 0.0 ^a^	100.0 ± 0.0 ^a^

Estimated marginal means denoted by different superscript letters within the same column represent statistically significant differences for the respective variable (*p* < 0.05), as determined by two-way ANOVA with Tukey’s HSD post hoc multiple comparison test.

**Table 4 foods-14-02700-t004:** Estimated marginal means of Se concentration and the percentage of true Se retention on the main effects of seafood species and cooking methods.

Common Name	Estimated Marginal Means ± Standard Error	
	Se (μg/100 g of Product)	TR (%)
Different species of seafood:		
Pacific white shrimp	33.4 ± 3.6 ^g^	84.2 ± 10.5 ^b,c^
Banana prawn	63.2 ± 10.5 ^d,e^	95.2 ± 8.3 ^a^
Giant Tiger Prawn	48.3 ± 11.2 ^e,f,g^	86.6 ± 17.1 ^b^
Ornate rock lobster	52.4 ± 10.6 ^e,f,g^	65.3 ± 19.2 ^d^
Musk Crab	69.6 ± 17.4 ^d,e^	74.9 ± 7.9 ^c,d^
Blue crab	68.2 ± 25.0 ^d,e^	82.5 ± 19.4 ^b,c^
Serrated Mud Crab	64.2 ± 20.5 ^d,e^	65.8 ± 25.4 ^d^
Red frog crab	12.1 ± 4.8 ^h^	87.8 ± 21.0 ^b^
Splendid squid	35.9 ± 4.8 ^g^	52.5 ± 11.1 ^e^
Cuttlefish	57.8 ± 7.7 ^e,f^	77.9 ± 9.8 ^c^
Bigfin reef squid	43.8 ± 9.9 ^f,g^	100.0 ± 0.0 ^a^
Razor clam	64.8 ± 6.4 ^d,e^	91.8 ± 6.1 ^a^
Oysters	70.3 ± 16.3 ^c,d,e^	70.9 ± 10.6 ^c,d^
Cockle	79.4 ± 32.2 ^c,d^	81.3 ± 17.1 ^b,c^
Clam	61.6 ± 16.7 ^d,e^	81.5 ± 5.8 ^b,c^
Mussels	80.5 ± 28.1 ^b,c^	92.5 ± 6.0 ^a^
Wedge shell	96.2 ± 32.1 ^b^	100.0 ± 0.0 ^a^
Indo-Pacific horseshoe crab	154.1 ± 35.9 ^a^	83.1 ± 16.0 ^b,c^
Northern whiting fish	55.7 ± 13.0 ^e,f^	97.3 ± 2.9 ^a^
Silver pomfret	69.7 ± 24.4 ^d,e^	100.0 ± 0.0 ^a^
Cooking technique in different species of seafood:		
Fresh	50.8 ± 27.8 ^d^	-
Boiling	54.8 ± 21.3 ^c^	79.7 ± 19.3 ^c^
Frying	78.8 ± 38.5 ^a^	88.8 ± 14.5 ^a^
Grilling	71.6 ± 32.8 ^b^	82.1 ± 15.1 ^b^

Values assigned different superscript letters within the same column, corresponding to either fish species or cooking methods, indicate statistically significant differences for the respective variable (*p* < 0.05), as determined by two-way ANOVA followed by Tukey’s HSD post hoc multiple comparison test.

## Data Availability

The original contributions presented in the study are included in the article, further inquiries can be directed to the corresponding author.

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
