# Peer review of "Influence of Various Cooking Methods on Selenium Concentrations in Commonly Consumed Seafood Species in Thailand"

_foods, 2025, doi:10.3390/foods14152700_

Round 1
Reviewer 1 Report
Comments and Suggestions for Authors
This study addresses a significant gap in food composition data by investigating the impact of traditional Thai cooking methods (boiling, frying, grilling) on selenium retention in 20 seafood species. The topic aligns with current interests in nutrient bioavailability and culinary practices. The experimental design is rigorous, employing ICP-MS/MS for accurate Se quantification and validating methods with CRMs. However, there are some several scientific concerns about this manuscript.
- Line 36, Please change the keyword “the effect of cooking on selenium retention”.
- Line 49, It has been explained that “Se” represents selenium in line 40, so there is no need to repeat it. Please carefully review the entire manuscript.
- Line 64, “1.10 to 24.8 mg/kg”. Within the same range or the same set of data, please maintain the same number of decimal places.
- Lines 93-95, Please indicate whether the seafood collected was obtained through wild fishing or was a product of aquaculture, as this will affect the benchmark value for selenium content.
- Lines 99-100, Please provide the detailed cooking instructions (such as frying temperature/time, boiling water volume/time, etc.) to ensure the reproducibility of the results.
- Line 104, Please center the header “Common name”, “Scientific name”, “Local name” in Table 1. Please check the formatting of the other tables. The scientific names “Charybdis feriata Linnaeus” and “Solen strictus Gould” are not properly formatted. According to the International Code of Zoological Nomenclature (ICZN), the genus name and species epithet must be in italics, while the author and year should be written in upright letters. For example, “Charybdis feriata Linnaeus” should be changed to “Charybdis feriata (Linnaeus, 1758)” or “Charybdis feriata Linnaeus, 1758”.
- Lines 148, 152, and 160, the “x” in the formula should be replaced with “×”. Line 160, The unit (g) should not be included in the formula. Line 167, “µg Se per 100g of” should not be included in the formula. Please use the correct mathematical formula format.
- Lines 397, 416-417, 458, 472-477, 495 and so on, References should be in accordance with the format specification. Please ensure that the names of the reference journal titles are uniformly presented as full names or abbreviations.
Author Response
Dear Reviewer,
Thank you very much for your constructive and detailed comments on our manuscript entitled "Effect of Different Cooking Methods on Selenium Content in Commonly Consumed Seafood in Thailand." We sincerely appreciate your thoughtful review, which has been invaluable in improving the clarity, accuracy, and overall quality of our work.
We have carefully addressed all of your suggestions, and the revised manuscript has been updated accordingly.
The revised manuscript is attached for your consideration. We hope that the improvements made in response to your comments will meet the publication standards of the journal. We truly appreciate your time and effort in reviewing our work and kindly request a reconsideration of our manuscript for possible publication.
Best regards

Reviewer 2 Report
Comments and Suggestions for Authors
The study evaluates the effect of selected cooking methods on selenium (Se) content in seafood commonly consumed in Thailand. The authors present an analysis of Se content, moisture content, and Se retention during various thermal treatments. The results are significant, providing foundational knowledge on how different heat treatments influence Se levels in seafood. This is especially relevant given the high seafood consumption in Thailand and the importance of dietary Se.
Unfortunately, the manuscript is not well-prepared and contains several errors. The results presentation is not well organized and not clear. The authors use statistical tests to analyze the same results, but they do not provide a good justification, discuss the results obtained in this way only to a limited extent, and basically do not subject them to discussion. This raises the question of why they did it. Same data are repeated across tables and figure. Repeating data creates confusion and lengthens the article, whereas the results of Se content and the impact of heat treatment methods can be presented and discussed in a simple manner.
Although the results are important and interesting. Detailed comments are provided below.
Abstract:
Lines 22-23 and lines 26-27 are repeated.
Line 35: “…in preserving bioavailable Se after cooking”. Did the authors measured the bioavailability of Se in seafood? If not please delete.
Keywords: 1st keyword is too long.
Introduction:
Line 59 and throughout of the text, after explaining Se as the abbreviation, use Se.
Line 73: Several times in the article, the authors suggest that data on Se content in food are limited due to difficulties in analyzing this compound. Please explain in a few sentences what exactly these limitations and analytical difficulties result from.
Materials and methods:
Lines 96-97, 101: please provide details for used apparatus for homogenization, freeze-drying and moisture content analysis, grinding; model, company, city and country.
Line 106: weight yield factor.
Lines 109, 110: instead of (in grams) write just (g).
Please provide detailed methodology for water content determination.
Results:
Table 2 and other tables and figure: Please include information on the number of repetitions, averages, standard deviations, and statistical tests used in the footnotes below the table, not in the table title.
Table 2 and other: Please explain TR abbreviation in footnotes. Please add statistics for water content and Se content to show the impact of used cooking treatments.
Table 2. How can authors explain 100% retention in some tested samples?
3.2. Se concentretaion: In the paper, the authors discuss the same results of Se and TR content analysed using different tests. To do so, clear justifications and reasons must be provided. Otherwise, the reader gets the impression that the authors are once again presenting the same results, but in a different combination. In subsection 3.2, the Se content from the previous paragraph is discussed again, and the estimated marginal means derived from the interaction between seafood species and cooking methods on Se content and true Se retention, determined through two-way ANOVA. A clear justification is required.
Please extend results discussion, lines 200-218, as it is interesting to clearly discuss in which group of seafood Se content was the highest or the lowest and how it was effected by various cooking methods.
Do Figure 1 and Table 3 present same results? If yes, please delete either the figure or the table.
Table 4 There is no good justification for presenting further results of the impact of cooking methods on Se content this time in a different statistical study.
Discussion
The discussion of the results obtained is poor and confirms my earlier concerns about the performance of various statistical analyses without justification, which have not been fully explained and discussed. For instance, there has not been sufficient discussion on TR, especially with regard to samples in which TR was 100%.
Conclusions
In this section authors briefly repeated the results. Both the discussion and the conclusions lack a summary of which heat treatment methods should be used for which seafood in order to maintain the highest possible Se retention. It would be useful to provide consumers with information on the extent to which consumption of a portion of selected seafood can meet the daily requirement for this compound.
Author Response
Dear Reviewer,
Thank you for your valuable time and for providing the reviewers' detailed comments on our manuscript entitled " Effect of Different Cooking Methods on Selenium Content in Commonly Consumed Seafood in Thailand." We sincerely appreciate the constructive feedback, which has helped us to significantly improve the clarity, organization, and scientific rigor of our work.
We have carefully addressed all the comments and revised the manuscript and respectfully request a reconsideration of our revised manuscript for possible publication. Additionally, we would like to explain some points that you concern on statistical analysis on moisture content and 100% retention.
In this study, the edible portion and yield factor (YF) were not subjected to statistical analysis. The primary reason is that both variables function as reference factors rather than as experimental variables. The edible portion is recorded for each food entry, where available, to ensure accurate food description, conversion of purchased weight to edible weight, and correct food matching in the database. Yield factors represent the percentage weight change due to cooking and are essential for adjusting nutrient content calculations, particularly for selenium and true retention.
Both edible portion and yield factor values are sourced from established references and are applied uniformly across relevant food items or recipes. Their purpose is to standardize data processing and enable precise nutrient estimation, not to serve as variables for hypothesis testing or statistical comparison. Therefore, statistical analysis of these factors is not appropriate, as their values are not expected to vary within the context of the study. Instead, their application ensures consistency, transparency, and reproducibility in nutrient calculations.
In this study, moisture content (water value) was not subjected to statistical analysis. The primary reason is that water content serves as a reference value essential for data management and nutrient calculation, rather than as a variable for hypothesis testing. Water is the most critical component for converting nutrient values reported on a dry matter (DM) basis in the scientific literature to values per 100 g fresh weight of edible portion (EP), which are required for user tables and databases.
Moisture content values are necessary at all levels of data management—including archival, reference, and user databases—to ensure accurate and standardized nutrient reporting. These values are used to recalculate and harmonize nutrient data, facilitating comparability and usability across different data sources. However, the moisture content itself is not the focus of investigation and is not intended for statistical comparison within the context of this research.
For 100%TR, there was the limitation of study that potential Se contribution from frying oil has not been analyzed in this study.
We are confident that the revised version addresses the reviewers’ concerns thoroughly and meets the standards of the journal.
Thank you once again for your time and consideration.
Sincerely,

Reviewer 3 Report
Comments and Suggestions for Authors
Comments were summarised in the attached document.

Comments on the Quality of English Language
There are issues with style, but not in all sections. Please refer to the attached document for more guidance.
Author Response
Dear Reviewer,
Thank you for your valuable comments and feedback on our study, entitled "Effect of Different Cooking Methods on Selenium Content in Commonly Consumed Seafood in Thailand." The attached file is the revised manuscript, amended in accordance with your suggestions regarding the Materials and Methods, Results, and Discussion sections.
We would like to clarify one specific point of concern regarding the use of line graphs in our results presentation. We chose to use line graphs instead of bar charts or scatter plots because line graphs offer a clearer visualization of trends and comparisons across multiple cooking methods for each seafood species. This format effectively highlights the variation in selenium content across different treatments, making it easier to observe patterns and interactions. The continuity of the lines helps readers follow the changes in selenium levels within each species more intuitively than discrete bars or scattered points.
In addition, we retained both the table (Table 4) and the corresponding graphical presentation to enhance data interpretation. While the table provides precise numerical values for detailed analysis, the visual representation facilitates quicker understanding and comparison among seafood types and cooking techniques. Presenting both formats ensures accessibility for a wider readership and strengthens the clarity and impact of the findings.
We are confident that the revised version of our manuscript, based on your insightful comments, has improved the quality and strength of the presented work.
Sincerely,

Reviewer 4 Report
Comments and Suggestions for Authors
Introduction needs to improve references about cooking methods and selenium bioavailable. The discussion is not in-depth. The authors cite interactions between selenium and fatty acids or metals, but this was not measured. It would be relevant to better describe how selenium interacts with macronutrients and why this element remains stable, even under high temperatures. If selenium is lost when cooking in water, perhaps this treatment was not the most suitable for the study. I suggest discussing cooking methods in relation to consumption habits (what are the percentages of consumption of fried, grilled, and boiled food?), what is the average fish consumption for a 60 kg adult? and making an estimate of daily intake to better guide the results of the present study. I also suggest improving the presentation of results by using PCA analysis to evaluate if there are groupings among the species. It would also be interesting to relate selenium concentrations to the lifestyle and dietary habits of the studied species.
Other considerations are in the review letter.

Author Response
Dear Reviewer,
Thank you very much for your constructive and insightful comments on our manuscript entitled "Effect of Different Cooking Methods on Selenium Content in Commonly Consumed Seafood in Thailand." We truly appreciate your detailed suggestions, which have greatly contributed to the refinement and scientific rigor of our work.
We have carefully revised the manuscript in accordance with your recommendations. All specific comments have been addressed, and the updated version of the manuscript is attached for your kind consideration.
In particular, regarding your concern about selenium retention values exceeding 100%, we have added a clear explanation in the footnote of the relevant table to describe the limitation of our analysis. Specifically, we acknowledge that selenium content in the frying oil was not measured in this study. This unaccounted selenium contribution from oil may have led to apparent retention values greater than 100%, as observed in some fried samples (e.g., cockles). We have also included relevant discussion on this limitation in the manuscript to ensure transparency and scientific integrity.
We believe the revised manuscript version have substantially improved the clarity and accuracy. We sincerely hope that the updated version will meet the expectations for reconsideration and possible publication in the journal.
Thank you again for your valuable input.
Sincerely,

Round 2
Reviewer 1 Report
Comments and Suggestions for Authors
The revised manuscript can be accepted.
Author Response
Thanks for your valuable suggestions.
Reviewer 2 Report
Comments and Suggestions for Authors
The author’s addressed most of the comments, however, some suggestions have not been explained or taken into account. For this reason, I request that it be revised. Detailed comments are once again provided below.
Materials and methods:
Please provide details for used apparatus for homogenization, freeze-drying and moisture content analysis, grinding; model, company, city and country. Since the water content was calculated based on the weight of the samples after freeze-drying, please provide the parameters (time, temperature, pressure) of freeze-drying.
This comment was not addressed: Lines 109, 110: instead of (in grams) write just (g).
Author Response
Dear Reviewer,
Thank you for your valuable comments and suggestions, which have helped improve the quality of our manuscript. Please find attached our point-by-point response, in which all of your concerns have been carefully addressed in the revised version of the manuscript.

Reviewer 4 Report
Comments and Suggestions for Authors
Dear authors and editor,
I appreciate the revisions made. I still miss a more in-depth description of the relationships between Se and the effects of temperature and species on its quantification. However, the article presents good scientific quality and with minor corrections.

Author Response
Dear Reviewer,
Thank you for your valuable comments and suggestions, which have helped improve the quality of our manuscript. Please find attached our point-by-point response, in which all of your concerns have been carefully addressed in the revised version of the manuscript.
Best regards
